# A Study on the Mechanisms Influencing Older Adults’ Willingness to Use Digital Displays in Museums from a Cognitive Age Perspective

**DOI:** 10.3390/bs14121187

**Published:** 2024-12-13

**Authors:** Anan Hu, Beiyue Chen, Sai Liu, Jin Zhang

**Affiliations:** 1Department of Tourism, Fudan University, Shanghai 200433, China; huanan@fudan.edu.cn (A.H.); jinz@fudan.edu.cn (J.Z.); 2Institute of Archaeology, University College London, London WC1E 6AE, UK; beiyue.chen.24@ucl.ac.uk

**Keywords:** older adults, digital exhibition technology, cognitive age, museum, technology acceptance model

## Abstract

As older adults age, changes in their physiological, psychological, and cognitive functions often lead to inherent anxiety and fear regarding the use of digital technologies. Cognitive age, reflecting an older adult’s mental perception of their chronological age, is a crucial moderating factor in shaping their willingness and behavior towards technology adoption. However, the mechanisms through which cognitive age impacts the behavior of older adults within the realm of digital technology utilization remain unclear. Thus, grounded in the Extended Technology Acceptance Model (ETAM) and employing structural equation modeling, this study intends to investigate mechanisms which influence older adults’ behavioral intentions towards the use of museum digital displays. Consequently, it confirms that attitudes mediate the correlation between perceived usefulness, subjective norm, perceived self-efficacy, and the behavioral intention towards museum digital display utilization. Perceived ease of use does not directly affect attitudes, but instead exerts an indirect impact on attitude through perceived usefulness. Cognitive age negatively moderates the relationship between attitudes and the behavioral intention to use digital technologies. Additionally, the mediating role of attitudes in the correlation between older adults’ perception of museum digital displays and behavioral intention is moderated by cognitive age. Specifically, older adults with a higher cognitive age value (who perceive themselves to be younger than their chronological age) exhibit a weaker mediating effect of attitudes on the relationship between perceived usefulness and behavioral intention compared to those with a lower cognitive age value (who perceive themselves to be older than their chronological age). The findings seek to unbox the “black box” of how cognitive age mediates the correlation between attitudes and behavioral intention towards the use of museum digital displays among older adults, providing valuable theoretical insights for the bidirectional enhancement of digital technology utilization, the overall well-being of older adults, and high-quality development in museums.

## 1. Introduction

Aging has been a global trend which has brought not only challenges to economic development, social stability, and welfare, but also new opportunities to the world. The World Population Prospects 2024 report by the United Nations Department of Economic and Social Affairs (UNDESA) predicts that global life expectancy may reach 77.4 years by 2054 and that by 2080, the number of people aged 65 and above will exceed the population aged 18 and below [1]. In an age where scientific innovation drives technological iteration and the development of digital technology, the “active aging” initiative by the World Health Organization promotes the use of digital technology terminals amongst older adults [2].

China, one of the fastest-aging countries in the world, is reported to have 21.1% of its population aged 60 and above by 2023, totaling 297 million. As the “active aging” initiative has been emphasized and promoted by the government in recent years, internet technology becomes a major way of older adults to pursue “lifelong learning”. According to *The 54th Statistical Report on China’s Internet Development*, published by the China Internet Network Information Center (CNNIC), as of June 2024, the number of internet users in China had reached nearly 1.1 billion, with new users primarily consisting of adolescents aged 10–19 and older adults. Notably, individuals aged 50–59 and those aged 60 and above respectively accounted for 15.2% and 20.8% of new internet users [3]. Additionally, data indicate that the proportion of older adults using the internet in China rose from 5.4% in 2017 to 14.3% in 2022 [4]. Recent reports reveal that more than half of Chinese seniors aged 65–69 now use smartphones, while 31.2% of those aged 70–79 are also smartphone users [5]. Despite this significant increase in internet and smartphone usage among older adults, disparities in their ability to explore and utilize advanced digital functions remain pronounced. Most older adults primarily use smartphones for basic activities such as online messaging, reading news, and watching videos. However, the adoption of more complex functions, such as mobile payments, online medical appointment bookings, ride-hailing services, and group buying for dining or entertainment activities, remains relatively low. These challenges are compounded by barriers such as limited technical familiarity and insufficient information accessibility. Many older adults report difficulties stemming from small interface icons, non-intuitive navigation, and overly complex procedures. These difficulties often lead to “fear of use” and “inability to use effectively”, slowing the older adults’ integration into the digital age. In response, China is actively exploring strategies to promote digital inclusion among the elderly. Initiatives such as age-friendly app adaptations, digital literacy workshops, and “care mode” settings aim to shift older adults from passive acceptance to proactive engagement with digital technologies. This trend of developing digital technologies and applications for older adults has good reason to be extended into museums.

Given the preference of older adults for informal learning outside traditional classrooms, museums, offering “varied experiences for education, enjoyment, reflection and knowledge sharing” and endeavoring to be “accessible and inclusive”, have in-born advantages and thus are ideal places to offer life-long education to older adults [6]. However, it is pointed out that the “majority of the museum exhibition programs and educational activities are predominantly tailored to the provision for the school age children, the young elder people, the general public … with very little attention to the elderly” [7], so that the potential of museums to provide lifelong learning has not been exploited to the full, and that the willingness of older adults to perceive digital technological features and use the digital facilities in museums has not been explored. Thus, it is high time that more attention be diverted to the older adult audience in museums.

Existing studies connected to digital technology utilization amongst older adults have focused on contextual examination [8,9,10], influencing factors [11], characteristics of use [12], and efficacy of use [13]. As far as the context of museums is concerned, most of the studies have centered on the characteristics of older adults’ use of technology [14] and the extension effects such as education [15]. Little is mentioned about the influence of the older adults’ perception of their digital display usage on their attitudes and willingness to use the technologies.

Moreover, cognitive age is a distinctive intergenerational factor reflecting the behavioral choices of contemporary older adults [16,17]. Previous research on the cognitive age of older adults has predominantly focused on areas such as health and consumer behavior, consistently highlighting its close connection with factors like living environment [18], workplace, working abilities [19], cognitive capabilities and disabilities [20], and other internal and external influences. Studies have shown that older consumers with varying cognitive ages display significant differences in personal values [21], service environment preferences [22], shopping behaviors [23], and aesthetic and cultural tastes [24]. However, research exploring the relationship between cognitive age and the acceptance of digital technologies remains limited. Existing studies primarily address challenges such as rejection of new technologies [25], stereotypes [26], and adaptation difficulties [27] stemming from differences in perceived cognitive age. Future research should aim to resolve the internal conflict of cognitive age perception among older adults by leveraging digital technology interventions to guide them from passive acceptance to active engagement. Encouraging proactive use of digital tools could significantly enhance their quality of life and overall sense of well-being. Through the previous literature review, we identify that the significance of cognitive age has been neglected in previous studies in the field of information technology usage for older adults. Therefore, we adopt the extended technology acceptance model (ETAM) to explore the influence of older adults’ perception of technological features of museum digital displays on their attitudes and willingness to use them, and the moderating role of their perceived age. Two research questions are raised, one being how older adults perceive the characteristics of museum digital displays andhow their perception affects their digital displayusage, and the other being what role cognitive age plays in affecting the older adults’ relationship with the digital display.

## 2. Literature Review and Hypotheses Development

### 2.1. Museum Digital Display

Museums have increasingly applied digital technology to exhibitions in recent decades, producing QR codes, audio guides, VR multimedia displays, AR assisted displays, immersive scene design, AI assistant self-service tours and online live guiding to vitalize the stories of exhibits and enrich the museum tour experience. Up until now, there has not been an agreed-upon holistic term for the digital forms of displays in museums. This study takes up the general term “digital display”, which refers to displays set in the milieu of museums, using computer technology to digitize the physical museum exhibits with the aim that the audience is free to choose what to see and how to see it according to their personal needs.

In the past, researchers primarily focused on specific forms and effects of digital display. For instance, Kajinami et al. investigated the digital display cases’ ability to inform visitors of background information [28], Li & Liew focused on the capability of the “interactive user interface” to enhance visitor experience in museums and art galleries [29], and Jung et al. delved into how visitors’ experiences in museums were affected by virtual reality and augmented reality regarding the escape experience [30]. The panorama of digital display usage has received little notice, not to mention the impact of older adult visitors’ perception of museum digital displays on their willingness and action. This study emphasizes the older adults’ experience and reception of museum digital displays.

### 2.2. Older Adults’ Use of Informational Technologies

Currently, the definitions of the term “older adult” vary in academia. The definition of “older adult” can be delineated into two scopes. Firstly, in a broad sense, “elderly“ literally means “people who have become old”. Under this definition, an “older adult” is featured by the deterioration of biological functions, psychological adaptability, social engagement, and economic capacity [31]. Secondly, from a narrower perspective where age is the criterion, older adults are defined by the UN as “persons over 65 years of age”, a definition adopted by many countries to set legal age standards. Taking a middle ground between the two definitions, this study refers to Mathur et al.‘s definition of old age which uses a “younger cut-off” of 55 years old [32]. There are two reasons for adopting such definition, as follows. To begin with, from the perspective of marketing, “older consumers” as a market segment is stipulated as people aged 55 years old and above [33]; furthermore, a number of previous studies related to senior tourism regard tourists over 55 years of age as senior tourists. In summary, in the museum tourism scenario, the age group between 50 to 59 will become the major potential consumers in the future tourism market. Therefore, the research population of this study is set to be senior tourists over 55 years old.

In addition, with the innovation, iteration, and popularization of information technology, more and more older adults have started to utilize information technologies in their life. Older adults’ information technology utilization can be mainly divided into general use (e-mail, social networks, etc.), information acquisition (information search, news reading, etc.), communication (e.g., online shopping), and so on. It has been demonstrated that using information technologies helps older adults to improve social participation, sustain and grow social networks, enhance living standards, and increase individual social capital [34]. However, at the same time, due to physiological and psychological changes caused by aging, there is a gap in information technology acceptance and utilization between older adults and young people. Perceived usefulness, perceived ease of use, perceived self-efficacy, expectation of results, and moderating factors such as age, socio-economic status, cultural factors, and experience in using information technology have become the main motivating factors affecting the use of information technology by older adults [35].

In the context of museums, the use of museum digital displays belongs to the general topic of information technology usage. Previous studies suggested that the older adults’ use of digital displays in museums is subject to various physical, psychological, and social factors [36]. On the one hand, the use of digital display technologies among older adults is associated with internal factors, including but not limited to age, level of education [37,38], physical health [39], social economic status [40], internet skills [41], motivation [42], self-efficacy [43], and sense of happiness [44]. On the other hand, older adults’ technology usage is affected by external factors. For instance, family support is discovered to contribute to the higher possibility of longer usage of technologies among older adults [45]. The atmosphere of tourism websites contributes to enhancing older tourists’ familiarity with online travel platforms [46]. In summary, the question of how to promote better use of digital displays by older adults to improve their museum experience, empowering the older adults with technology usage ability, and fulfilling their entertainment needs, has become the core issue on which this study focuses.

### 2.3. Technology Acceptance Model (TAM) and Social Cognitive Theory (SCT)

The theories commonly used in research regarding users’ behavior in the adoption of informational technologies include the Theory of Reasoned Action (TRA), Technology Acceptance Model (TAM) and its variations (TAM2, TAM3, UTAUT), Social Cognitive Theory (SCT), and Social-emotional Selectivity Theory (SST).

Taking into account the characteristics of information technology utilization within older adults and the ability of museum digital display to augment reality and create virtuality, in order to construct a succinct but comprehensive model to explore the dynamic mechanisms of older adults’ perception of museum digital display on their behavioral intentions of actual use, we decided to select crucial factors from two of the most frequently used models in previous researches, TAM and SCT, from which many of the above-mentioned factors influencing older adults’ information technology use derive.

The Technology Acceptance Model (TAM), introduced by Davis and adapted from the Theory of Reasoned Action (TRA), is tailored for modeling user acceptance of information systems [47]. TAM postulates that perceived usefulness (PU) and perceived ease of use (PEU) directly influence the attitude towards using a certain technology (AU), and hence affect behavioral intention (BI). Furthermore, PEU affects PU. PU denotes the user’s subjective likelihood that employing a specific application system will enhance job performance within an organizational context [48]. PU is conceptualized as the probability that older adults think the digital display can improve their visiting experience in this study. Attitude, reflecting an individual’s opinion and evaluation of a particular object or phenomenon, is directly shaped by older adults’ preference for digital displays [49]. Previous researches have confirmed that perceived usefulness has a positive effect on the attitude and intention to use the technology [50], and this positive effect applies to all age groups, including older adults [51].

PEU in TAM is referred to as the degree to which a user expects the target system to be easy to use [52]; PEU, in this study, is defined as the extent to which the older adults think the digital display device is easy enough to use. Compared with young people, older adults have lower learning speed and poorer technology mastery, and are thus more inclined to accept easy-to-use technologies. As shown in previous studies, PEU exerts positive influence on the attitude and behavioral intention to use digital technologies [53]. This facilitating effect is more prominent for older adults than for younger people [54]. In short, based on the TAM, PEU, and PU in the study are older adults’ subjective perception and evaluation of the digital displays, which both influence their attitude towards the museum digital display utilization.

Thus, in this endeavor, the following hypotheses are proffered:

**Hypothesis 1 (H1).** 
*Perceived usefulness (PU) positively impacts attitude (AU) towards utilization of digital displays.*


**Hypothesis 2 (H2).** 
*Perceived ease of use (PEU) positively impacts attitude (AU) towards utilization of digital displays.*


**Hypothesis 3 (H3).** 
*Perceived ease of use (PEU) positively impacts perceived usefulness (PU).*


**Hypothesis 4 (H4).** 
*Attitude (AU) towards using digital displays positively impacts behavioral intention (BI).*


The Technology Acceptance Model 2 (TAM2), proposed by Venkatesh in 2003, adds social influence processes and cognitive instrumental processes to the original TAM model [55]. One of the most important factors added is the “Subjective norm” (SN), which is referred to as “a person’s perception that most people who are important to him think he should or should not perform the behavior in question” [56]. Within the realm of museums, SN is defined as the older adult visitors’ belief that people who are important to him, including their family, friends, colleagues, etc. think they should or should not use digital display devices. The subjective norm, a conceptual factor in social psychology, reflects the extent to which social pressures influences older adults’ adoption of museum digital displays [57]. The closer the relationship between these social groups and older adults, the greater their impact on the latter’s ambivalence and behavioral intention towards engaging with museum digital displays. As older adults age, they have stronger emotional needs, a most important one of which being the need to belong to a group [58]. It has been confirmed that older people use information technology mainly to get news and new knowledge, to facilitate their contact with friends and relatives, and to develop more shared topics with young people, ultimately strengthening their sense of belonging and maintaining a positive connection with society. The desire to obtain a sense of belonging in a way affects older adults’ attitudes and willingness to use the museum digital displays [55]. SN is an objective factor, external to the older adults themselves, indicating the effect of the environment in shaping elderly visitors’ attitudes towards digital displays. Therefore, this study proposes the following hypotheses:

**Hypothesis 5 (H5).** 
*Subjective norm (SN) positively impacts attitude (AU).*


The Social Cognitive Theory (SCT), put forward in 1986 by Bandura posits that behavioral change is made possible by a personal sense of control [59]. The key constructs of SCT involve perceived self-efficacy (PSE), outcome expectancies, goals and socio-structural impediments and facilitators. PSE is referred to as “people’s belief in their capabilities to perform an action required to attain desired outcome”, often identified as an important factor influencing older adults’ information system use [60]. In this study, different from PU and PEU, which focus on the subjective perception of the devices, PSE emphasizes individual’s assessment of their competence to utilize digital displays [61]. Older adults with high perceived self-efficacy often exhibit a positive attitude towards challenging activities, such as exploring new technologies like museum digital displays. Compared to those with low perceived self-efficacy, they tend to demonstrate a stronger interest and preference for digital displays. It has been found that the perceived self-efficacy of older people in using information technology is significantly lower than that of younger people [51]. The low self-efficacy leads to technology anxiety and negative emotions in using museum digital displays, and hence undermining the attitude and willingness to use [62]. On the contrary, high perceived self-efficacy can promote information technology use, and external encouragement and support can enhance the perceived self-efficacy of older adults in using museum digital displays [55].

Consequently, in this endeavor, the following hypotheses are proffered:

**Hypothesis 6 (H6).** 
*Perceived self-efficacy (PSE) positively impacts attitude towards utilization of digital displays (AU).*


### 2.4. The Moderating Role of Cognitive Age (CA)

This study discovers a factor that is often neglected by previous research—cognitive age (CA). CA is the age individuals perceive themselves to be and is defined by four dimensions: feel age, look age, do age, and interest age [63]. Cognitive age, as a key representation of older adults’ self-concept, refers to their subjective evaluation of cognitive abilities relative to their chronological age [31]. It encompasses perceptions of self-assessed cognitive capacity, learning efficiency, memory, and problem-solving skills compared to their actual age. As such, cognitive age is regarded as a critical factor for understanding and predicting older adults’ health status and their receptiveness to new products and technologies. According to Barak and Schiffman [64], when it comes to the study of the attitudes and behavioral intentions of the elderly group, CA is a more accurate predictor than physical age. Schiffman and Sherman theorizes that the elderly with young CA ‘‘perceive themselves as younger in age and outlook, … feel more self-confident and in control of their lives” [65]. In other words, the older adults who perceive themselves as younger than their actual age are more positive in their life attitudes and behaviors. Regarding the influence of cognitive age in technology use, some researchers have used cognitive age as an independent variable; for instance, Eastman and Iyer (2005) found that seniors with young cognitive age use the internet more than those with old cognitive age, and other scholars have applied cognitive age as a moderator in their models [66]. According to Chaouali and Souidenin, in their research on factors influencing older adults’ resistance to online banking systems, CA moderates the relationships between psychological and functional barriers as well as between functional barriers and mobile banking resistance [59]. It was discovered that the elderly with who perceive themselves to be older than their chronological age are more hampered by the barriers which will lead to resistant behaviors than their counterparts with young CA. Hong et al. verified the moderating effect of CA between the PU, PEU, perceived enjoyment, SN and behavioral intention of older adults to use mobile data services [67]. They discovered that for the elderly “young at heart”, the positive effect of PU, PEU, and perceived enjoyment on IT acceptance decisions is positively moderated by young cognitive age. Kim et al. also identified that CA moderates the relationship between website atmosphere and website familiarity, and younger CA is observed to enhance revisiting intentions [46].

What is worthy of further attention is that according to Haug et al., the tendency of older adults to feel that they are younger than they really are is especially strong, even stronger than that of younger adults [68]. Particularly in the museum setting, whether the museum digital display is a timely assistant or an “icing on the cake” for older adults of different cognitive age, and whether the functional value or the hedonic value comes first in older adults’ perception of digital displays are questions that have attracted our attention. Therefore, in order to avoid an overgeneralization of older adults as inactive and resistant to information technologies, we add the factor CA between attitude and behavioral intention to examine whether, in the context of museum exhibitions, CA plays a moderating role, and how. Thus, the study proposes the following hypotheses:

**Hypothesis 7 (H7).** 
*Cognitive age (CA) positively moderates the correlation of attitude and behavioral intention (BI)—the younger the cognitive age is (the higher the CA value of the questionnaire items), the more enhanced is the positive relationship between attitude towards using the digital display (AU) and the behavioral intention of using digital display (BI).*


**Hypothesis 8 (H8).** 
*The mediating role of attitudes is moderated by cognitive age. PU, SN, PSE, and PEU influence BI through AU to a greater extent in older adults with older cognitive age compared to older adults with younger cognitive age (older adults who have higher CA value).*


Based on the literature review and study hypothesis above, the conceptual framework has been delineated and graphically displayed in Figure 1.

## 3. Methodology

### 3.1. Questionnaire Design

The measurement questionnaire used in this work was a scale containing 30 items. The questionnaire can be divided into two parts—demographic information and a scale for the seven factors of the model. The demographic characteristic section of the questionnaire includes items such as gender, age, education level, occupation (before retirement), and monthly income. The scale section examines the interrelationships among the seven factors of proposed model: perceived usefulness (PU), perceived ease of use (PEU), subjective norm (SN), perceived self-efficacy (PSE), attitude (AU), cognitive age (CA), and behavioral intention (BI). All the factors are measured by the 5-point Likert scale (1 and 5 demonstrate “strongly disagree” and “strongly agree”). The items of PU and PEU derive from Davis [47]; the items used to measure SN were adapted from Ajzen [56] and Taylor and Todd [69]; the items of PSE were adapted from Schwarzer [70]; the items of CA were adapted from Barak and Schiffman [64]; the items of AU were adapted from Fishbein and Ajzen [71]; the items of BI were adapted from Limayem and Chin [72]. With regard to the details provided in Appendix A, the factor definitions and measurements utilized were derived from prior research, which was further adapted to the specific research background.

### 3.2. Data Collection

We adopted a random sampling method to distribute and collect questionnaires in this study. Before the formal research, 50 questionnaires were distributed to elderly respondents with experience in using museum digital displays through snowball sampling for pre-survey testing. In the aim to assess the clarity, measurability, and scientific accuracy of questionnaire items, the respondents were asked to complete the survey anonymously. The results showed that 50 valid questionnaires were recovered from the pre-survey, and that the reliability of the questionnaire was significant. Therefore, a further formal survey could be carried out. The official questionnaires were distributed offline to older adults above 55 in Shanghai, Xi’an, and Nanjing from 10 June to 18 June 2020. A total of 337 questionnaires were distributed; excluding 34 invalid questionnaires, 303 valid questionnaires were received, which could be used for the research of this paper (Table 1).

### 3.3. Data Analysis

The study employed 6 data analysis techniques, including demographic and descriptive analysis, structural equation modeling, reliability and validity analysis, regression analysis, correlation analysis, and structural equation modeling (SEM). SPSS25.0 and Smart PLS 3 were used to conduct statistic and analyze. This study first tested common method bias (CMB) in the data to test multicollinearity between constructs, then implemented the two-step analysis method proposed by Anderson and Gerbing. Specifically, a CFA was initially conducted on a theoretical measurement model followed by testing hypothetical relationships after obtaining a satisfactory construct model [73]. Additionally, the study evaluated the indirect effects of attitude through the bootstrapping method.

## 4. Results

To test the main effect (H1–H6), the moderated relationship (H7) and the proposed mediated relationship, we used the structural equation modeling (SEM) technique and a moderated regression analysis with SPSS (version 24), respectively. The moderated mediation relationship was tested with Hayes PROCESS macro [74].

### 4.1. Measurement Model

Reliability and validity tests are prerequisites for discriminating the validity of a questionnaire. In this study, the reliability of the items was measured by the Cronbach alpha coefficient of the constructs, and the convergent and discriminant validity were measured to assess the structural validity of the questionnaire. As shown in Table 1, the values of composite reliability (CR) ranged from 0.841 to 0.897, indicating a high level of internal consistency for all latent variables [75]. Standardized loadings of all constructs exceeded the threshold value of *p* < 0.05, and the average variance extracted (AVE) of each construct was greater than the criterion value of 0.50. These results suggest satisfactory convergent validity [76]. In addition, discriminant validity was first examined with correlation coefficients of the constructs (Table 2). Correlations among all constructs were under 0.50, indicating four distinct constructs in the model. The AVE of each underlying construct (from 0.55 to 0.86) was higher than squared correlations between the construct and other study constructs (from 0.00 to 0.14), demonstrating discriminant validity.

Meanwhile, the measurement model was evaluated through CFA, the results of which demonstrated that the model fit is good: χ^2^⁄df = 1.105, RMSEA = 0.019, NFI = 0.956, RFI = 0.946, IFI = 0.996, TLI = 0.995, CFI = 0.996, GFI = 0.941, AGFI = 0.921.

### 4.2. Structural Model and Hypotheses Testing (H1–H6)

The structural model was scrutinized and assessed using SPSS, which exhibited a robust model fit, with the exception of verification towards “perceived ease of use--attitude”: χ^2^⁄df = 1.326, RMSEA = 0.033, NFI = 0.953, RFI = 0.943, IFI = 0.988, TLI = 0.985, CFI = 0.988, GFI = 0.937, AGFI = 0.916. The parameter estimates (Table 3) indicated that perceived ease of use is significantly, positively related to perceived usefulness (β = 0.76, *p* < 0.01); perceived usefulness is also significantly, positively related to attitude (β = 0.27, *p* < 0.01); subjective norm is significantly, positively related to attitude, too (β = 0.20, *p* < 0.01); perceived self-efficacy is significantly, positively related to attitude (β = 0.64, *p* < 0.01); and attitude is significantly, positively related to behavioral intention (β = 0.79, *p* < 0.01). These results support H1, H3, H4, H5, and H6. However, one path does not pass the test: perceived ease of use is not related to attitude (β = −0.21, *p* > 0.01). Thus, H2 is invalid. There are mainly two reasons behind this result. On the one hand, aging triggers a chain reaction on physiological, psychological and cognitive levels, including visual and auditory disorders, lack of mobility, weakened self-confidence, decreased attention, and memory. The older adults’ perception of the ease of use of museum digital displays is manifested as technical barrier and resistance psychology, which leads to the insignificant impact on the attitude towards use. On the other hand, the museum digital display often requires the additional use of Wechat and other external applications to scan QR codes and to visualize the background information of the exhibits. The cumbersome prerequisite procedures add obstacles for older adults to perceive that the ease of use of the technology is tantamount to clumsiness, and the double technical anxiety and fear (of both the display itself and additional use of smartphone software) leads to the result that the perceived ease of use does not directly affect the attitude of using museum digital displays, but instead indirectly affect the attitude through the mediating role of perceived usefulness.

### 4.3. Mediating Role of Attitude Between PU, PEU, SN, PSE, and BI

The study employs models 4 and 6 of the Bootstrapping method to examine how AU affects BI among older adults to use digital exhibits in museum exhibitions. A total of 5000 repetitions were set, followed by the examination of the confidence intervals (CI) for the indirect effects. If the 95% CI does not encompass 0 and the non-standardized measurement values are significant at the *p* < 0.05 level, then the existence of a mediating effect can be verified. As in Table 4, AU exhibits significant mediating effects in the relationships between PU, SN, PSE, PEU to PU, and BI, validating that AU mediates the influence of the older adults’ perception of museum digital displays on their behavioral intention. PEU affects BI through a chain-mediated pathway of PU and AU.

### 4.4. Moderating Effect of CA (H7)

The study employs model 14 of PROCESS to examine the moderating effect of CA on the relationship between AU and BI among the older adult population, and the bootstrap resampling procedure was conducted 5000 times to obtain the test results. The results are presented in Table 5 and Figure 2. The structural model showed a good model fit: χ^2^⁄df = 1.295, RMR = 0.024, RMSEA = 0.031, NFI = 0.935, RFI = 0.924, IFI = 0.984, TLI = 0.982, CFI = 0.984, GFI = 0.920, AGFI = 0.899. The interaction term between CA and AU significantly and negatively influences the older adults’ BI to use museum digital display (β = −0.094, *p* < 0.05), suggesting that CA exerts a negative moderating effect on the direct connection between AU and BI, thus subverting Hypothesis 7. This surprising finding overturns our built-in assumption that younger-minded older adults (higher in CA value) have a higher level of adaptability and acceptance of digital technologies which helps to enhance the transformative process from attitude to behavioral intention to use digital technology. As a matter of fact, cognitive age inhibits the transition from attitude to behavioral intention. This finding reflects the status quo that the generic museum digital displays designed based on the built-in “commonsensical” cognitive model of older adults often fails to engageolder adults.

Firstly, regarding the interface design of museum digital displays, the “age-neutral” universal interface often involves complex multi-layered menu operations and standard specifications for icons and font sizes. Interactive elements, such as AI-guided navigation, typically require detailed commands to activate and operate effectively. These highly “digitally intelligent” design features can significantly affect older adults with lower cognitive age values, leading to a sense of difficulty and reducing the accessibility of digital displays. For those with higher cognitive age values, the perceived complexity and unintuitive nature of these technologies may exceed their expectations and interest, resulting in a negative moderating effect of cognitive age.

Furthermore, in terms of content presentation, the digital exhibits often rely on straightforward descriptions of artifacts and academic narratives, which increase the cognitive load for older adults with lower cognitive age values. Simultaneously, such declarative interpretative methods fail to meet the needs of those with higher cognitive age values to expand their knowledge, which further amplifies the negative moderating effect of cognitive age on attitude and behavioral intention.

As described above, the negative moderating effect of cognitive age is not an insurmountable barrier to older adults’ adoption of museum digital displays. External interventions can mitigate or compensate for this negative impact. For instance, training and guidance could provide older adults with lower cognitive age values with basic skills such as touch screen navigation, page turning, and zooming. For those with higher cognitive age values, advanced training could include online artifact comparison and simulated artifact restoration, encouraging the transition from positive attitudes towards digital exhibitions to actionable behavioral intentions.

Additionally, social support interventions, such as launching simplified digital display apps and mini-programs designed for older adults, organizing museum family open days where family and relatives assist seniors in using digital displays, and fostering online communities for peer-to-peer digital skills sharing, can help bridge the digital divide among older adults.

Thus, while cognitive age plays a negative moderating role, digital inclusion, differentiated design, and targeted interventions can empower older adults with varying cognitive age values, enhance their acceptance and recognition of museum digital display, and ultimately increase their willingness use the digital displays.

### 4.5. Mediated Moderation

Following the approach outlined by Edwards et al. for testing moderated mediation, it examines the mediating role of AU in correlation between factors of museum digital displays perceived by the older adults (PU, PEU, SN, PSE) and BI under the moderation of CA [77]. Considering different levels of CA, Table 6 displays moderated mediation testing results of AU. CA negatively moderates the mediating effect of AU in correlations between PU (β_Difference_ = −0.020, *p* < 0.01), SN (β_Difference_ = −0.017, *p* < 0.01), PSE (β_Difference_ = −0.019, *p* < 0.01), PEU to PU (β_Difference_ = −0.017, *p* < 0.01), and BI of the older adult population. In other words, for older adults with varying CA levels, significant differences exist in the indirect effects of PU, SN, PSE, and PEU to PU on BI through the mediation of AU. For older adults with young CA, the influences of PU, SN, PSE, and PEU to PU on BI via AU are relatively weaker than those of the older adults with old CA.

As discussed above, this study, through an exploration of the factors influencing older adults’ willingness to use museum digital displays, has revealed several surprising findings. Firstly, older adults’ perceived ease of use of museum digital displays does not directly impact their attitudes but rather exerts an indirect influence through the mediating effect of perceived usefulness. Secondly, cognitive age inhibits the transition from older adults’ attitudes towards museum digital displays to their behavioral intentions. Thirdly, for older adults who perceive themselves as younger than their actual age, the influence of perceived usefulness (PU), subjective norm (SN), perceived self-efficacy (PSE), and the ease-of-use-to-usefulness relationship on behavioral intentions (BI) through attitudes (AU) is relatively weaker compared to those whose perceived cognitive age aligns more closely with their chronological age.

## 5. Conclusions, Discussion, and Implications

### 5.1. Conclusions and Theoretical Contributions

Considering older adults, this study delves into mechanisms about how their perception of museum digital displays impacts their behavioral intention to use digital displays. The following conclusions are drawn:

#### 5.1.1. Mediating Role of AU

Through investigating the transmission mechanism between the older adults’ perceptual factors of museum digital displays on their BI and the mediating effect of AU, the study finds that the older adults’ perception of museum digital displays exerts an indirect positive influence on BI through the mediation of AU. Specifically, AU plays a mediating role in correlation between PU, SN, PSE, PEU to PU, and BI to digital display utilization among older adults. In addition, PEU positively affects BI through the chain mediation of PU and AU. This implies that AU, as an embodiment of emotional value in the perception of museum digital displays, plays a significant role in facilitating and enhancing the willingness of the elderly to use digital displays. Therefore, this study can be viewed as an extension and supplementation to the TAM model, when tailored to museums and the elderly population [78]. Meanwhile, the study also reinforces the positive mediating effect of positive attitude on digital technology acceptance and utilization in exhibitions, as already shown in the original TAM model.

#### 5.1.2. Moderating Effect of CA

Cognitive age, one of the differentiating characteristic variables reflecting individual differences, has been confirmed in previous research to play a role in the activities and behaviors among the older adult population [59], but little is mentioned about the moderating effect of CA on the correlation between AU and BI to use digital technologies. It is of great significance in constructing a theoretical framework to illuminate the relationship between AU and BI. As is shown in the moderating effect analysis results, CA exhibits a negative moderating effect (with a path coefficient of −0.094) on path coefficient correlation between AU and BI to utilize museum digital displays among older adults. The result of this study contradicts the original hypothesis and our common-sense perception of the positive acceptance of museum digital displays by older adults with a younger mentality. In the real-life scenario of museums, older adults with high cognitive age value—i.e., older people who perceive they are psychologically younger than their actual age—only use digital displays as an auxiliary exhibition tool. The functional value of digital displays is far from enough to satisfy older people’s self-guided museum experience, which deviates from the traditional conception of the cognitive structure of older adults’ utilization of digital technologies.

This finding suggests that scenarios in which digital technologies are applied can affect users’ behavioral intention. At the same time, the finding also points out that the research, development, and promotion of museum digital displays should comprehensively consider and respect the behavioral preferences of different age groups. The user’s experience should be taken into consideration in addition to the way the content is presented. Digital displays should be highlighted as an organic part in the experiential process of digital technologies, rather than a means of technological assistance for exhibitions. The provision of suitable museum digital display content and services for different age groups will become the future direction of research. Thus, this study provides practical guidance for previous theoretical research on digital technology application and acceptance, and also enriches the current academic field in broadening the research scenarios of digital technology application.

#### 5.1.3. Mediated Moderation Effect

It is confirmed that the indirect relationship between the perception of museum digital displays and BI among older adults to utilize the displays is mediated by AU and moderated by CA. Specifically, for older adults with young CA, the weaker is the indirect relationship mediated by AU between the perceived factors of museum digital displays and BI, the path coefficient being smaller than that of older adults with old cognitive age. In other words, the older adults with a younger mindset exhibit a better acceptance of the easy-to-use functional and emotional value of technology; however, the general digital displays in museums, with cumbersome operation programs are contrary to the needs of the older adults, affecting BI towards digital displays directly. Consequently, they require less mediation through AU for the transmission of the positive effects of the perception of museum digital displays on BI. Conversely, for older adults with old cognitive age, the indirect relationship between the perception of museum digital displays and BI through the mediation of attitudes is stronger. On one hand, older adults with old cognitive age are more inclined to recognize the difficulty of accepting digital technologies due to old age, thus influencing their BI to use digital displays through affecting AU. On the other hand, these older adults, characterized by high identity consistency, small individual differences, and a low value perception of the functional value of digital displays, rely more on the indirect effect of AU to transmit the positive impact of the perception of museum digital displays on their BI [79].

### 5.2. Suggestion

Having analyzed the factors contributing to older adults’ digital display usage in museums, the study has reached conclusions that bear significance to the future improvement of museum digital displays. Advice will be given from two aspects: digital display design and museum management.

#### 5.2.1. Digital Display Design

When designing digital displays, emphasis should be placed on perceived usefulness and ease of use, there is a favorable correlation between these two perceptions and behavioral intention to engage with digital displays.

In order to enhance the perceived usefulness towards digital designs, designers must ensure that these designs significantly contribute to the museum visiting experience, whether they operate as standalone exhibits or are integrated with other designated exhibits. To achieve this objective, the form and content of the digital display itself should be well integrated, while the form and content of any specific digital display should be consistent with the design style and the storyline of the whole exhibition. Apart from improvements on the existing means of display, digital designers should develop new forms of digital designs to expand functionality and enhance usefulness, meeting the somatogenic and cognitive needs of older adult visitors.

To enhance the perceived ease of use, which is a factor indirectly influencing attitude and behavioral intention through its impact on perceived usefulness, designers must address the challenges that older adults encounter while using digital displays. Three approaches can be undertaken. First, functions and operational procedures should be simplified, with a reduction in the number of functional layers to prevent confusion among the older adults. Second, proper education and guidance should be provided prior to display use. Special attention should be given to addressing possible cognitive challenges. By combining simple language with vivid illustrations in the user education section of the display, the operational procedures will be more easily comprehensible to the older adults. Third, the incorporation of additional electronic terminals should be avoided. Using digital displays creates difficulties to older adults, and additional QR codes or smartphone mini-apps attached to the displays will do no good but put an extra burden on older adults during their museum trips.

#### 5.2.2. Museum Management

Museums possess significant potential in fostering the use of digital displays and creating an environment conducive to their utilization. Museum administrators should focus particularly on promotional strategy and positioning of digital displays, as well as older adult visitor research and evaluation.

To begin with, tracing everything back to the root, museums should make clear the positioning of designing digital displays and contemplate how older adult visitors, with their unique perceptual characteristics, should be received. The digital displays should be an integral part of the exhibition experience rather than auxiliary facilities designed solely for specific age groups. The museums should boost the cooperation among the curatorial team, the digitalization team, and even the education team, so that the curatorial team can contribute to the content of digital displays, the education team can contribute to the communication and education strategies of the digital displays, and, finally, the design team can put everything together to make digital displays that can be seamlessly incorporated into the exhibition.

As population aging is becoming increasingly severe, museums should raise their awareness of older adult visitors. While these older adult visitors should be given special attention and assistance, efforts should be made to avoid alienating them from other visitors. What’s more, it is best to avoid over-generalization among the older adults, since individual differences in cognitive age render a starkly contradictory moderating effect, as is shown in the statistics above. Functions designed according to the characteristics of older adults, both physical and mental, can be broken up and re-integrated into different digital displays so that the older adults will feel the ease and usefulness of the digital displays without feeling that they are the vulnerable population who needs special care, thus reducing the negative moderating effect of cognitive age.

Furthermore, to counter the negative moderating effect of cognitive age and to enhance the positive effect of the subjective norm on the older adults, the museum can create an environment for older adults to share their experience of digital displays, so that, influenced by their peers, those older adults who are previously intimidated by, ignorant of, or uninterested in digital displays can be encouraged to try the digital displays out. For instance, museums can hold offline sharing sessions and online discussion forums where older adults are free to chat to each other about their engagement with digital displays, the expected procedures of usage and what they have learned from the digital displays.

Last, but not least, only with deep and extensive visitor research and evaluation can museums obtain an accurate “digital display user portrait” of older adults, and make improvements specific to their needs. According to various studies, although an increasing number of museum practitioners in China acknowledge the importance of visitor research and evaluation, these studies and evaluations are rarely employed in museum practices [80], let alone studies especially concerning older adult visitors. Museum audience researchers should cater to a diverse range of older adults with different characteristics—cognitive age being one—and design the questions accordingly. The audience research should be conducted both online and offline, to avoid excluding the older adults who have difficulty accessing internet technologies. To achieve in-depth understanding of the expectations, needs, and usage outcomes of older adults, the museums should focus on all the pre-visit, during-visit, and after-visit experiences of older adults with digital displays. To facilitate the collection of samples among the older adults, the museums can provide small incentives, such as discounts or free passes, to encourage older adults’ participation.

### 5.3. Limitations and Further Direction

There exist some limitations, as this study acknowledges, primarily manifested in sample selection, the choice of museum digital display format, the refinement of variable queries, and the efficacy of the survey.

#### 5.3.1. Sample Selection

By now, China, while having the largest older adult population in the world, also has the most complex digital technology application scenarios, making itself a most important case destination for exploring the practices of digital technology for older adults. The questionnaires in this study were exclusively distributed offline, primarily in Shanghai, Xi’an, and Nanjing. Since museums have the clustered distribution and a high degree of digitalization in these destinations, the respondents are relatively familiar with museums and digital displays. However, such a high level of familiarity among older adults should only be cautiously generalized due to the constraint in sample selection. Future investigations may consider supplementary surveys in more remote or less culturally saturated regions to conduct in-depth research and complement this limitation.

#### 5.3.2. Choice of Museum Digital Display Format

This study investigated digital display experiences by using Shanghai Museum’s 3D imaging exhibits as a simple example. However, in reality, museums employ diverse forms of digital displays, some with complex systems or interfaces and others with intricate content. Consequently, the impact of selecting different digital exhibition formats on relevant outcomes remains a subject for further exploration. Subsequent research may categorize display formats to delve into the effects on different operational complexities, content, and functions of the use of digital displays by older adult visitors.

#### 5.3.3. Refinement of Variable Queries

In order to facilitate responses from older adults, the four questions regarding cognitive age were measured with a Likert 5-point scale. However, this approach lacks precision. Future research may consider employing cognitive age measures using segmented age scales for a more nuanced evaluation.

#### 5.3.4. Survey Efficacy

The current study is a static cross-sectional analysis. Future research may adopt a dynamic process-oriented perspective to observe the learning and acceptance of the older adult population towards digital displays. Further comparative analyses are poised for a more holistic comprehension on older adult population’s reception of digital displays, facilitating the proposal of targeted improvement measures, providing insights for the development of museum digital displays.

## Figures and Tables

**Figure 1 behavsci-14-01187-f001:**
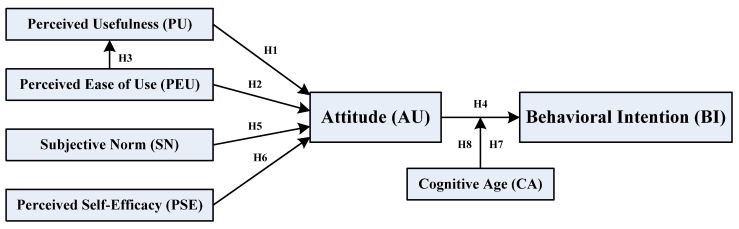
Hypothetical model.

**Figure 2 behavsci-14-01187-f002:**
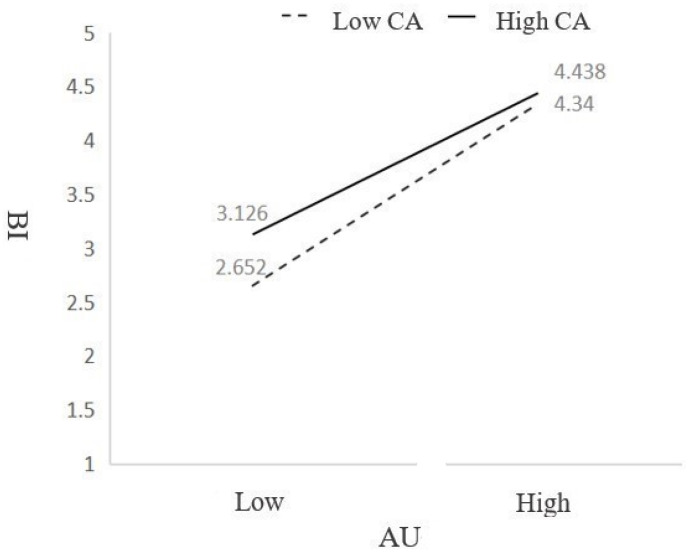
Moderating effects of CA.

**Table 1 behavsci-14-01187-t001:** Demographic characteristics of older adult respondents (N = 303).

Variables		Frequency (%)
Gender	Male	40.26
	Female	59.74
Age	55–65	75.25
	65–75	20.79
	≥75	3.96
Work state	Working	17.49
	Retired	82.51
Education level	Middle school or below	41.25
	High school/Special school/Technical school	48.84
	Two-year college/Four-year university	9.91
Monthly Income	≤8000	95.71
	8001–16,000	4.29
Occupation (current/before retirement)	Public officials	4.62
	Enterprise employees	59.74
	Self-employed	9.24
	Professionals (e.g., teachers, doctors, etc.)	5.28
	Other	21.12

**Table 2 behavsci-14-01187-t002:** Questionnaire constructs, reliability, and validity.

Constructs	Cronbach’s α	Factor Loading	CR	AVE	1	2	3	4	5	6	7
1.PU	0.889	0.851–0.855	0.889	0.728	0.853						
2.PEU	0.865	0.770–0.801	0.866	0.617	0.767	0.786					
3.SN	0.941	0.830–0.864	0.887	0.724	0.532	0.608	0.851				
4.PSE	0.867	0.802–0.843	0.866	0.684	0.661	0.783	0.684	0.827			
5.AU	0.859	0.781–0.827	0.861	0.674	0.626	0.647	0.677	0.819	0.821		
6.BI	0.841	0.784–0.818	0.841	0.638	0.692	0.625	0.664	0.773	0.793	0.799	
7.CA	0.895	0.809–0.894	0.897	0.743	0.397	0.404	0.456	0.502	0.443	0.507	0.862

Note: The diagonal line is the square root of the AVE, and the numbers below the diagonal line are the correlation coefficients among constructs.

**Table 3 behavsci-14-01187-t003:** Hypothesis examination results.

Hypothesis	Hypothesis Path	Path Coefficient	Std. Error	t-Value	*p*-Value	Testing Results
H1	PU → AU	0.274	0.077	3.564	***	pass
H2	PEU → AU	−0.212	0.111	−1.911	0.056	Not pass
H3	PEU → PU	0.768	0.063	12.224	***	pass
H4	AU → BI	0.793	0.061	12.984	***	pass
H5	SN → AU	0.208	0.057	3.675	***	pass
H6	PSE → AU	0.64	0.095	6.737	***	pass

Note: *** *p* < 0.001.

**Table 4 behavsci-14-01187-t004:** Mediation testing results.

Path	Std. Error	Effect	Bias-Corrected 95% CI			Percentile 95% CI		
			Lower	Upper	*p*	Lower	Upper	*p*
PU → AU → BI	0.073	0.235	0.103	0.391	***	0.105	0.392	***
SN → AU → BI	0.053	0.199	0.091	0.301	***	0.094	0.303	***
PSE → AU → BI	0.09	0.587	0.427	0.779	***	0.427	0.779	***
PEU → PU → AU → BI	0.06	0.184	0.081	0.318	***	0.081	0.318	***

Note: *** *p* < 0.001.

**Table 5 behavsci-14-01187-t005:** Moderating examination results.

Hypothesis Path	Path Coefficient	Std. Error	t-Value	*p*-Value	Testing Results
CA × AU → BI	−0.094	0.058	−2.380	0.017	Not Pass

**Table 6 behavsci-14-01187-t006:** Moderated mediation testing results.

Path	Std. Error	Effect	Bias-Corrected 95% CI			Percentile 95% CI		
			Lower	Upper	*p*	Lower	Upper	*p*
std_high_AGI	0.059	0.156	0.061	0.293	0	0.057	0.286	0.001
std_low_AGI	0.079	0.231	0.094	0.404	0.001	0.095	0.405	0.001
std_high_CGI	0.043	0.138	0.061	0.232	0.001	0.056	0.225	0.002
std_low_CGI	0.06	0.204	0.086	0.318	0.003	0.091	0.323	0.002
std_high_DGI	0.091	0.411	0.257	0.614	0	0.25	0.599	0
std_low_DGI	0.11	0.608	0.425	0.855	0	0.427	0.859	0
std_high_BAGI	0.048	0.122	0.046	0.236	0	0.044	0.229	0.001
std_low_BAGI	0.065	0.181	0.073	0.329	0.001	0.073	0.332	0.001

Note: std_high: moderated by high cognitive age; std_low: moderated by low cognitive age. AGI: PU → AU → BI; CGI: SN → AU → BI; DGI: PSE → AU → BI; BAGI: PEU → PU → AU → BI.

## Data Availability

Data are contained within the article.

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
