# Peer review of "A Study on the Mechanisms Influencing Older Adults’ Willingness to Use Digital Displays in Museums from a Cognitive Age Perspective"

_behavsci, 2024, doi:10.3390/bs14121187_

Round 1
Reviewer 1 Report
Comments and Suggestions for Authors
1. Introduction section reflects generally situation in China, and can not be generalized to a wider sample, so obviously authors need to present more details about the routine of use of gadgets by Chinese older adults.
2. The description of methods is too brief and unclear. How many items were used for each factor? Were they constructed by authors or taken from some valid instument?
3. It would be reasonable to mention the number of the model/s used from Hayez Process, so that this study could be replicated by other researchers.
Comments on the Quality of English LanguageMany phrases are unclear, starting from the abstract, some sentences lack meaningful parts that affect understanding of the text.
Author Response
Comment 1:
Introduction section reflects generally situation in China, and can not be generalized to a wider sample, so obviously authors need to present more details about the routine of use of gadgets by Chinese older adults.
Response 1:
Thank you for pointing this out. We agree to this comment. In our previous manuscript, we did not provide sufficient details about the general practices of older adults in China regarding the use of electronic products which provides a panoramic view specific to China. We sincerely apologize for this oversight. Therefore, in the "Introduction" section, we have addressed this issue in greater detail, with revisions highlighted in red (Lines45–71). Specifically, we have incorporated official statistical data to present an overview of internet and electronic product usage among older adults in China. Additionally, we have included detailed descriptions of the specific practices and contexts in which older adults use digital products. These revisions aim to provide a more comprehensive representation of the overall situation regarding electronic product and digital technology usage among older adults in China.
China, one of the fastest-aging countries in the world, is reported to have 21.1% of its population aged 60 and above by 2023, totaling 297 million. As “active aging” initiative is emphasized and promoted by the government in recent years, the internet technology becomes a major way of older adults to pursue “lifelong learning”. According to The 54th Statistical Report on the China’s Internet Development published by the China Internet Network Information Center (CNNIC), as of June 2024, the number of internet users in China had reached nearly 1.1 billion, with new users primarily consisting of adolescents aged 10–19 and older adults. Notably, individuals aged 50–59 and those aged 60 and above respectively accounted for 15.2% and 20.8% of new internet users[3].Additionally, data indicate that the proportion of older adults using the internet in China rose from 5.4% in 2017 to 14.3% in 2022 [4]. Recent reports reveal that more than half of Chinese seniors aged 65–69 now use smartphones, while 31.2% of those aged 70–79 are also smartphone users[5]. Despite this significant increase in internet and smartphone usage among older adults, disparities in their ability to explore and utilize advanced digital functions remain pronounced. Most older adults primarily use smartphones for basic activities such as online messaging, reading news, and watching videos. However, the adoption of more complex functions, such as mobile payments, online medical appointment bookings, ride-hailing services, and group buying for dining or entertainment, remains relatively low. These challenges are compounded by barriers such as limited technical familiarity and insufficient information accessibility. Many older adults report difficulties stemming from small interface icons, non-intuitive navigation, and overly complex procedures, often leading to "fear of use" and "inability to use effectively", thus slowing their integration into the digital age. In response, China is actively exploring strategies to promote digital inclusion among the elderly. Initiatives such as age-friendly app adaptations, digital literacy workshops, and "care mode" settings aim to shift older adults from passive acceptance to proactive engagement with digital technologies. This trend of developing digital technologies and applications for older adults has good reason to be extended into museums.
Comment 2:
The description of methods is too brief and unclear. How many items were used for each factor? Were they constructed by authors or taken from some valid instrument?
Response 2:
Thank you for highlighting this point. We agree with your comment. In our previous manuscript, we did not clearly specify the number of items used for each factor in the questionnaire design or the specific measurement methods in the "Methods" section. Accordingly, we have provided a detailed explanation of this aspect in the "Methods" section, with the revisions highlighted in red (Lines331–333, Appendix A).
With regard to the details provided in Appendix A, the variable definitions and meas-urements utilized were derived from prior research, which was further adapted to the specific research background.
Appendix A
Table A1. Items of Perceived usefulness, perceived ease of use, subjective norm, self-efficacy, attitude, behavioral intention and Cognitive age.
|
Construct |
No. |
Item |
|
Perceived usefulness (PU) |
1 |
Using museum digital display allows me to better understand exhibits |
|
2 |
Using museum digital display allows me to understand exhibits more quickly |
|
|
3 |
Using museum digital display enriches my understanding of exhibits |
|
|
Perceived ease of use (PEU) |
4 |
It does not take me a long time to learn how to use the functions of the museum digital display |
|
5 |
The operations of the museum digital display are clear and easy to understand for me. |
|
|
6 |
It is easy for me to learn and use the museum digital display skillfully |
|
|
7 |
I think the museum digital display functions are easy for me to use |
|
|
Subjective norm (SN) |
8 |
My family and relatives think I should use the museum digital display function |
|
9 |
My friends think I should use museum digital display function |
|
|
10 |
My colleagues (former colleagues) think I should use the museum digital display function |
|
|
11 |
Those whose opinions I value think I should use the museum digital display function |
|
|
12 |
People who have influenced me think I should use the museum digital display function |
|
|
13 |
Important people around me think I should use the museum digital display function |
|
|
Self-efficacy (SE) |
14 |
I have the ability to use and operate museum digital display |
|
15 |
I can quickly learn to use the museum digital display even without help |
|
|
16 |
I find it natural and satisfying to be able to operate the museum digital display skillfully |
|
|
Attitude (AU) |
17 |
I am pleased to use the digital display function in museum |
|
18 |
I enjoyed using the digital display in museum |
|
|
19 |
I am very like to use the digital display in museum |
|
|
Behavioral intention (BI) |
20 |
I would like to continue using the museum's digital display |
|
21 |
I hope the museum digital display can be applied in more scenarios comprehensively |
|
|
22 |
I hope the museum digital display can be more functional fully |
|
|
Cognitive age (CA) |
23 |
I feel younger than my actual age |
|
24 |
I look younger than my actual age |
|
|
25 |
I act younger than my actual age |
Comment 3:
It would be reasonable to mention the number of the model/s used from Hayez Process, so that this study could be replicated by other researchers.
Response 3:
Thank you for pointing this out. We agree with your comment. We have updated the sections “4.3. Mediating Role of Attitude between PU, PEU, SN, PSE, and BI” and “4.4. Moderating Effect of CA (H7)” to explicitly mention the model(s) used from Hayes' PROCESS. These revisions have been made in response to your suggestion and are highlighted in red in the manuscript (Line410-411, Lines422–424).
The study employs the model 4 and 6 of the Bootstrapping method to investigate how AU affect BI among older adults to use digital exhibits in museum exhibitions.
The study employs the model 14 of PROCESS to examine the moderating effect of CA on the relationship between AU and BI among the older adult population, and the bootstrap resampling procedure was conducted 5,000 times to obtain the test results.
/
Some other revisions have also been made to improve the manuscript. Brief descriptions of all the revisions are as follows:
List of Actions
LOA1: Introduction have been re-edited and re-written (line36-82, line90-105).
LOA2: The language, including the wording and grammar, of the whole essay has been polished (highlighted in red in the manuscript).
LOA3: The references have been updated (line 710-873).
LOA4: The Suggestion part has been enriched (Lines608–613, Lines619–631, Lines637–649).
Reviewer 2 Report
Comments and Suggestions for Authors
1. Content Contextualization
Comments: The manuscript thoroughly discusses the adoption of digital technologies by older adults in museum settings. However, some connections to broader theoretical frameworks and recent empirical studies in related fields are not fully explored. The discussion of previous studies could be more comprehensive to enhance the contextualization of the findings.
Suggestion: Consider integrating additional references to recent studies on cognitive age and digital acceptance among older populations outside the museum context to strengthen the background.
2. Research Design, Questions, Hypotheses, and Methods
Comments: The research design is robust, leveraging the extended Technology Acceptance Model (ETAM) and structural equation modeling. Hypotheses are clearly articulated and supported by a well-structured conceptual framework.
Suggestion: Ensure that all constructs in the model, such as subjective norm and cognitive age, are defined with clear operational definitions early in the text.
3. Arguments and Discussion of Findings
Comments: While findings are generally compelling, some sections lack a nuanced interpretation of the negative moderating effect of cognitive age. The discussion on practical implications could be further elaborated.
Suggestion: Enhance the discussion by linking findings to potential interventions or designs in digital displays that account for cognitive differences among older adults.
4. Results Presentation
Comments: The results are presented with sufficient detail, including statistical values and model fit indices. Tables are clear and well-organized.
Suggestion: Briefly summarize key findings at the end of the results section to reinforce their importance before moving to the discussion.
5. Referencing
Comments: While the manuscript is adequately referenced, some cited studies are relatively dated. Including more recent studies will add to the relevance and robustness of the manuscript.
Suggestion: Update the literature review with studies published in the last 3–5 years to reflect the latest trends in digital adoption among older adults.
6. Conclusions and Support from Results
Comments: Conclusions are logical but could better highlight how the results contribute to theory and practice. The implications for museum management and digital display design are insightful but could be more specific.
Suggestion: Strengthen conclusions by outlining clear, actionable recommendations for practitioners based on the findings.
Author Response
Comment 1:
Content Contextualization
Comments: The manuscript thoroughly discusses the adoption of digital technologies by older adults in museum settings. However, some connections to broader theoretical frameworks and recent empirical studies in related fields are not fully explored. The discussion of previous studies could be more comprehensive to enhance the contextualization of the findings.
Suggestion: Consider integrating additional references to recent studies on cognitive age and digital acceptance among older populations outside the museum context to strengthen the background.
Response 1:
Thank you for pointing this out. We agree with your comment. We have updated the manuscript in the section of “Introduction” adding the recent studies on cognitive age and digital acceptance among older adults outside the museum. These revisions have been made in response to your suggestion and are highlighted in red (Line90-105).
Moreover, cognitive age, a distinctive intergenerational factor reflecting the behavioral choices of contemporary older adults [16,17], Previous research on the cognitive age of older adults has predominantly focused on areas such as health and consumer behavior, consistently highlighting its close connection with factors like living environment[18], workplace, working abilities[19], cognitive capabilities and disabilities[20] and other internal and external influences. Studies have shown that older consumers with varying cognitive ages display significant differences in personal values[21], service environment preferences[22], shopping behaviors[23], and aesthetic and cultural tastes[24]. However, research exploring the relationship between cognitive age and the acceptance of digital technologies remains limited. Existing studies primarily address challenges such as rejection of new technologies[25], stereotypes[26], and adaptation difficulties[27] stemming from differences of perceived cognitive age. Future research should aim to resolve the internal conflict of cognitive age perception among older adults by leveraging digital technology interventions to guide them from passive acceptance to active engagement. Encouraging proactive use of digital tools could significantly enhance their quality of life and overall sense of well-being.
- Luo, C.; Yuan, R.; Mao, B.; Liu, Q.; Wang, W.; He, Y. Technology Acceptance of Socially Assistive Robots Among Older Adults and the Factors Influencing It: A Meta-Analysis. J. Appl. Gerontol. 2024, 43, 115-128, doi:10.1177/07334648231202669.
- Alexandrakis, D.; Chorianopoulos, K.; Tselios, N. Older Adults and Web 2.0 Storytelling Technologies: Probing the Technology Acceptance Model through an Age-related Perspective. Int. J. Hum.-Comput. Interact. 2020, 36, 1623-1635, doi:10.1080/10447318.2020.1768673.
- Soloveva, M.V.; Poudel, G.; Barnett, A.; Shaw, J.E.; Martino, E.; Knibbs, L.D.; Anstey, K.J.; Cerin, E. Characteristics of urban neighbourhood environments and cognitive age in mid-age and older adults. Health Place 2023, 83, 8, doi:10.1016/j.healthplace.2023.103077.
- Bonzini, M.; Comotti, A.; Fattori, A.; Serra, D.; Laurino, M.; Mastorci, F.; Bufano, P.; Ciocan, C.; Ferrari, L.; Bollati, V.; et al. Promoting health and productivity among ageing workers: a longitudinal study on work ability, biological and cognitive age in modern workplaces (PROAGEING study). BMC Public Health 2023, 23, 7, doi:10.1186/s12889-023-16010-1.
- Yu, J.H.; Ng, T.K.S.; Mahendran, R. Cognitive and physical age gaps in relation to mild cognitive impairment and behavioral phenotypes. GeroScience 2024, 46, 1129-1140, doi:10.1007/s11357-023-00864-9.
- Hofmeister-Tóth, A.; Neulinger, A. The importance and realization of personal values and cognitive age. Mark.-Trz. 2022, 34, 25-40, doi:10.22598/mt/2022.34.1.25.
- Kuppelwieser, V.G.; Klaus, P. Revisiting the Age Construct: Implications for Service Research. J. Serv. Res. 2021, 24, 372-389, doi:10.1177/1094670520975138.
- Rahman, O.; Yu, H. Key antecedents to the shopping behaviours and preferences of aging consumers A qualitative study. J. Fash. Mark. Manag. 2019, 23, 193-208, doi:10.1108/jfmm-12-2018-0165.
- McFarlane, A.; Samsioe, E. #50+fashion Instagram influencers: cognitive age and aesthetic digital labours. J. Fash. Mark. Manag. 2020, 24, 399-413, doi:10.1108/jfmm-08-2019-0177.
- Wang, W.H.; Zhang, Y.T.; Zhao, J.J. Technological or social? Influencing factors and mechanisms of the psychological digital divide in rural Chinese elderly. Technol. Soc. 2023, 74, 11, doi:10.1016/j.techsoc.2023.102307.
- Zhao, L.; Fu, B. Assessing the Impact of Recommendation Novelty on Older Consumers: Older Does Not Always Mean the Avoidance of Innovative Products. Behav. Sci. 2024, 14, 23, doi:10.3390/bs14060473.
- Cheng, X.S.; Qiao, L.Y.; Yang, B.; Li, Z.K. An investigation on the influencing factors of elderly people's intention to use financial AI customer service. Internet Res. 2024, 34, 690-717, doi:10.1108/intr-06-2022-0402.
Comment 2:
Research Design, Questions, Hypotheses, and Methods
Comments: The research design is robust, leveraging the extended Technology Acceptance Model (ETAM) and structural equation modeling. Hypotheses are clearly articulated and supported by a well-structured conceptual framework.
Suggestion: Ensure that all constructs in the model, such as subjective norm and cognitive age, are defined with clear operational definitions early in the text.
Response 2:
Thank you for pointing this out. We agree with your comment. In response, we have updated the manuscript in the sections “2.3. Technology Acceptance Model (TAM) and Social Cognitive Theory (SCT)” and “2.4. Technology Acceptance Model (TAM) and Social Cognitive Theory (SCT)” to include clear operational definitions for the constructs you mentioned. Specifically, we have supplemented the definitions of AU, SN, PSE, and CA. The revisions are highlighted in red in the manuscript (Lines199–200, 204–206, 229–233, 249-255 and 268–274).
Attitude, reflecting an individual’s opinion and evaluation of a particular object or phenomenon is directly shaped by older adults’ preference for digital displays [49].
PEU in TAM is referred to as the degree to which a user expects the target system to be effortless to use [52], PEU in this study is defined as the extent to which the older adults think the digital display device is easy enough to use.
Subjective norm, a conceptual factor in social psychology, reflects the extent to which social pressures influence older adults' adoption of museum digital display[56]. The closer the relationship between these social groups and older adults, the greater their impact on the latter's ambivalence and behavioral intention toward engaging with museum digital displays.
In this study, different from PU and PEU which focuses on the subjective perception of the devices, PSE emphasizes individual’s assessment of their competence to utilize digital displays [60]. Older adults with high perceived self-efficacy often exhibit a positive attitude toward challenging activities, such as exploring new technologies like museum digital display. Compared to those with low perceived self-efficacy, they tend to demonstrate a stronger interest and preference for digital displays.
Cognitive age, as a key representation of older adults' self-concept, refers to their subjective evaluation of cognitive abilities relative to their chronological age[31]. It encompasses perceptions of self-assessed cognitive capacity, learning efficiency, memory, and problem-solving skills compared to their actual age. As such, cognitive age is regarded as a critical factor for understanding and predicting older adults' health status and their receptiveness to new products and technologies.
Comment 3:
Arguments and Discussion of Findings
Comments: While findings are generally compelling, some sections lack a nuanced interpretation of the negative moderating effect of cognitive age. The discussion on practical implications could be further elaborated.
Suggestion: Enhance the discussion by linking findings to potential interventions or designs in digital displays that account for cognitive differences among older adults.
Response 3:
Thank you for highlighting this point. We agree with your comment. In the section “4.4. Moderating Effect of CA (H7),” we have enhanced the discussion by linking the findings to potential interventions or designs in museum digital displays that address cognitive differences among older adults. Specifically, we have explored how potential interventions in digital displays could be integrated with our findings to address the "black box" of the moderating effect of cognitive age. The revisions are highlighted in red in the manuscript (Lines 438–449).
First, regarding the interface design of museum digital display, the "age-neutral" universal interface often involves complex multi-layered menu operations and standard specifications for icons and font sizes. Interactive elements, such as AI-guided navigation, typically require detailed commands to activate and operate effectively. These highly "digitally intelligent" design features can significantly affect older adults with lower cognitive age values, leading to a sense of difficulty and reducing the accessibility of digital displays. For those with higher cognitive age values, the perceived complexity and unintuitive nature of these technologies may exceed their expectations and interest, resulting in a negative moderating effect of cognitive age.
Second, in terms of content presentation, the digital exhibits often rely on straightforward descriptions of artifacts and academic narratives, which increase the cognitive load for older adults with lower cognitive age values. Simultaneously, such declarative interpretative methods fail to meet the needs of those with higher cognitive age values to expand their knowledge, which further amplifies the negative moderating effect of cognitive age on attitude and behavioral intention.
As described above, the negative moderating effect of cognitive age is not an insurmountable barrier to older adults' adoption of museum digital displays. External interventions can mitigate or compensate for this negative impact. For instance, training and guiding could provide older adults with lower cognitive age values with basic skills such as touch screen navigation, page turning, and zooming. For those with higher cognitive age values, advanced training could include online artifact comparison and simulated artifact restoration, encouraging the transition from positive attitudes toward digital exhibitions to actionable behavioral intentions.
Additionally, social support interventions, such as launching simplified digital display apps and mini-programs designed for older adults, organizing museum family open days where family and relatives assist seniors in using digital displays, and fostering online communities for peer-to-peer digital skills sharing, can help bridge the digital divide among older adults.
Thus, while cognitive age plays a negative moderating role, digital inclusion, differentiated design, and targeted interventions can empower older adults with varying cognitive age values, enhance their acceptance and recognition of museum digital display, and ultimately increase their willingness use the digital displays.
Comment 4:
Results Presentation
Comments: The results are presented with sufficient detail, including statistical values and model fit indices. Tables are clear and well-organized.
Suggestion: Briefly summarize key findings at the end of the results section to reinforce their importance before moving to the discussion.
Response 4:
Thank you for highlighting this point. We agree with your comment. In the section “4.5. Mediated Moderation”, we have summarized the key findings briefly at the end of the results section to fortify their importance. The revisions are highlighted in red in the manuscript (Lines489–499).
As discussed above, this study, through an exploration of the factors influencing older adults' willingness to use museum digital display, has revealed several surprising findings. Firstly, older adults' perceived ease of use of museum digital display does not directly impact their attitudes but rather exerts an indirect influence through the mediating effect of perceived usefulness. Secondly, cognitive age inhibits the transition from older adults’ attitudes toward museum digital displays to their behavioral intentions. Thirdly, for older adults who perceive themselves as younger than their actual age, the influence of perceived usefulness (PU), subjective norms (SN), perceived self-efficacy (PSE), and the ease-of-use-to-usefulness relationship on behavioral intentions (BI) through attitudes (AU) is relatively weaker compared to those whose perceived cognitive age aligns more closely with their chronological age.
Comment 5:
Referencing
Comments: While the manuscript is adequately referenced, some cited studies are relatively dated. Including more recent studies will add to the relevance and robustness of the manuscript.
Suggestion: Update the literature review with studies published in the last 3–5 years to reflect the latest trends in digital adoption among older adults.
Response 5:
Thank you for pointing this out. We agree with your comment. In response, we have updated the manuscript in the sections “2.2. Older Adults’ Use of Informational Technologies”. We have added to the literature review the studies published in the last 3–5 years to reflect the latest trends in digital adoption among older adults. The revisions are highlighted in red in the manuscript (Lines168–175).
On the one hand, the using of digital display technologies among older adults is associated with internal factors, including but not limited to age, level of education [37,38], physical health [39], social economic status [40], internet skills [41], motivation [42], self-efficacy [43] and sense of happiness [44]. On the other hand, older adults’ technology usage is affected by external factors. For instance, family support is discovered to contribute to the higher possibility of longer usage of technologies among older adults [45]. The atmosphere of tourism websites contributes to enhancing older tourists' familiarity with online travel platforms [46].
Comment 6:
Conclusions and Support from Results
Comments: Conclusions are logical but could better highlight how the results contribute to theory and practice. The implications for museum management and digital display design are insightful but could be more specific.
Suggestion: Strengthen conclusions by outlining clear, actionable recommendations for practitioners based on the findings.
Response 6:
Thank you for pointing this out. We agree with your comment. In response, we have updated the manuscript in the sections “5.2. Suggestion” to strengthen conclusions by outlining clear, applicable recommendations for practitioners and the museum as a whole based on the findings. The revisions are highlighted in red in the manuscript (Lines608–613, Lines619–631, Lines637–649).
The museums should boost the cooperation among the curatorial team, the digitalization team and even the education team, so that the curatorial team can contribute to the content of digital displays, the education team can contribute to the communication and education strategies of the digital displays, and finally the design team can put everything together to make digital displays that can be seamlessly incorporated into the exhibition.
Functions designed according to the characteristics of older adults, both physical and mental, can be broke up and re-integrated into different digital displays so that the older adults will feel the ease and usefulness of the digital displays without feeling that they are the vulnerable population who needs special care, thus reducing the negative moderating effect of cognitive age.
Furthermore, to counter the negative moderating effect of cognitive age and to enhance the positive effect of subjective norm on the older adults, the museum can create the environment for older adults to share their experience of digital displays, so that influenced by their peers, those older adults who are previously intimidated by, ignorant of, or uninterested in digital displays can be encouraged to try the digital displays out. For instance, museums can hold offline sharing sessions and online discussion forums where older adults are free to chat to each other about their engagement with digital displays, the expected procedures of usage and what they have learnt from the digital displays.
Museum audience research should cater to a diverse range of older adults with different characteristics, cognitive age being one, and design the questions accordingly. The audience research should be conducted both online and offline, to avoid excluding the older adults who have difficulty accessing internet technologies. To achieve in-depth understanding of the expectation, needs and usage outcome of older adults, the museums should focus on all the pre-visit, during-visit and after-visit experience of older adults with digital displays. To facilitate the collection of samples among the older adults, the museums can provide small incentives, such as discounts or free passes, to encourage older adults participation.
/
Some other revisions have also been made to improve the manuscript. Brief descriptions of all the revisions are as follows:
List of Actions
LOA1: Introduction have been re-edited and re-written (line36-82, line90-105).
LOA2: The language, including the wording and grammar, of the whole essay has been polished (highlighted in red in the manuscript).
LOA3: The references have been updated (line 710-873).
LOA4: The Suggestion part has been enriched (Lines608–613, Lines619–631, Lines637–649).
